# Blocking HSV-1 glycoprotein K binding to signal peptide peptidase reduces virus infectivity *in vitro* and *in vivo*

**Shaohui Wang**, **Ujjaldeep Jaggi**, **Jack Yu**, **Homayon Ghiasi** *

Center for Neurobiology & Vaccine Development, Ophthalmology Research, Department of Surgery, Cedars-Sinai Medical Center, Los Angeles, California, United States of America

* ghiasih@cshs.org

**Data Availability Statement:** All relevant data are within the manuscript.

**Funding:** This research was supported by a National Eye Institutes (NIH-NEI) grant EY013615 (HG). The funders had no role in study design, data

## Abstract

HSV glycoprotein K (gK) is an essential herpes protein that contributes to enhancement of eye disease. We previously reported that gK binds to signal peptide peptidase (SPP) and that depletion of SPP reduces HSV-1 infectivity *in vivo*. To determine the therapeutic potential of blocking gK binding to SPP on virus infectivity and pathogenicity, we mapped the gK binding site for SPP to a 15mer peptide within the amino-terminus of gK. This 15mer peptide reduced infectivity of three different virus strains *in vitro* as determined by plaque assay, FACS, and RT-PCR. Similarly, the 15mer peptide reduced ocular virus replication in both BALB/c and C57BL/6 mice and also reduced levels of latency and exhaustion markers in infected mice when compared with control treated mice. Addition of the gK-15mer peptide also increased the survival of infected mice when compared with control mice. These results suggest that blocking gK binding to SPP using gK peptide may have therapeutic potential in treating HSV-1-associated infection.

## Author summary

Signal peptide peptidase (SPP) and HSV-1 glycoprotein K (gK) are essential genes in the host and virus, respectively. SPP and gK genes are both highly conserved. Previously we reported that gK binding to SPP is important for virus infectivity *in vitro* and *in vivo*. In this study we have identified the gK binding site to SPP and have shown that a gK peptide that blocks gK binding to SPP can block HSV-1 infectivity *in vitro* and *in vivo* using different strains of virus and mice. Thus, the ability of this peptide to block gK binding to SPP may be a useful tool to control HSV-1-induced eye disease in patients with herpes stromal keratitis (HSK).

## Introduction

HSV-1-induced corneal scarring (CS), broadly referred to as herpes stromal keratitis (HSK), is the leading cause of infectious blindness in developed countries [1,2]. While the precise viral

collection and analysis, decision to publish, or preparation of the manuscript.

**Competing interests:** The authors have declared that no competing interests exist.

pathogenicity determinant in ocular disease remains to be elucidated, we have shown that mice immunized with gK, but not with any other known HSV-1 glycoprotein, displayed significantly exacerbated CS, facial dermatitis, and blindness following ocular HSV-1 infection [3–6] and that these pathologies occur independent of the virus or mouse strain [5]. To further analyze the role of gK in CS, we constructed a recombinant HSV-1 that expresses two additional copies of the gK gene [7]. This recombinant virus significantly exacerbated CS in two different mouse strains compared to WT virus. We demonstrated that gK binds to signal peptide peptidase (SPP) and that this binding is essential for HSV-1 infectivity *in vitro* and *in vivo* [8–10]. Interestingly, binding of gK to HSV-1 UL20 is required for cell surface expression of gK [11–14]. Binding of the Golgi-specific DHHC zinc finger protein (GODZ) to HSV-1 UL20 was also required for proper cell surface expression of gK [14] and affected HSV-1 infectivity as shown using GODZ dominant-negative mutants, shRNA against GODZ, and GODZ knockout mice [13,14].

Our published studies using different combinations of mouse and virus strains with different recombinant viruses, knockout mice, and transgenic mice show the importance of gK in HSV-1-induced eye disease. gK is a highly hydrophobic protein with approximately 84% amino acid homology between HSV-1 and HSV-2 [15–17]. gK is a highly conserved gene and our previous results showed that blocking the interaction of gK with SPP reduces virus infectivity *in vitro* and *in vivo* [8,13]. Here we combined different gK fragments and gK peptides to fine map the gK binding site to SPP to 15 amino acids of gK.

We also sought to determine whether blocking this interaction may be an effective way to control HSV-1 infection *in vivo*. We found that this gK 15mer corresponding to the binding site of gK to SPP significantly reduced virus replication *in vitro* using three different HSV-1 strains. Application of this peptide as an eye drop reduced HSV-1 infectivity *in vivo* in both BALB/c and C57BL/6 mice and also increased survival in infected mice without altering cellular gene expression during primary infection. Finally, we have shown that this peptide reduced latency and eye disease in ocularly infected mice.

These studies are of great clinical significance because blocking the interaction of gK with SPP by targeted therapeutics might modulate and attenuate the immune response in HSV-1-induced eye disease. Further, control of corneal disease in HSV-1-challenged mice by using the gK peptide strongly suggests that blocking these interactions will be a clinically effective approach to reduce eye disease.

## Results

### Fine mapping of the gK region that binds to SPP

Previously we showed that full-length gK binds to SPP and that this binding is essential for HSV-1 infectivity *in vitro* and *in vivo* [8–10,18,19]. To determine what region of gK binds to SPP, we constructed three gK fragments corresponding to full-length gK as follows: gK1 (aa 1–115), gK2 (aa 116–230), and gK3 (aa 231–338) (Fig 1A). These plasmids were used to map the region of gK that binds to SPP by co-transfecting HeLa cells with each gK fragment and SPP plasmid as we described previously [9,14]. Control HeLa cells were transfected with vector DNA (empty plasmid). We found that gK1 was immunoprecipitated by anti-HA-SPP antibody in cells transfected with Flag-gK1 and HA-SPP (Fig 1B, gK1, upper blot). In contrast, gK2 and gK3 did not IP, indicating that they did not bind to SPP. To further fine map the region of gK1 that binds to SPP, we constructed an additional gK fragment (plasmid gK1.1) containing 85 amino acids (aa 31–115) of gK1 that excludes the gK signal sequence (aa 1–30) (Fig 1A, gK1.1). gK1.1 plasmid was co-transfected with SPP plasmid as above and similar to gK1, the gK1.1 fragment was immunoprecipitated by anti-HA-SPP antibody in cells transfected with

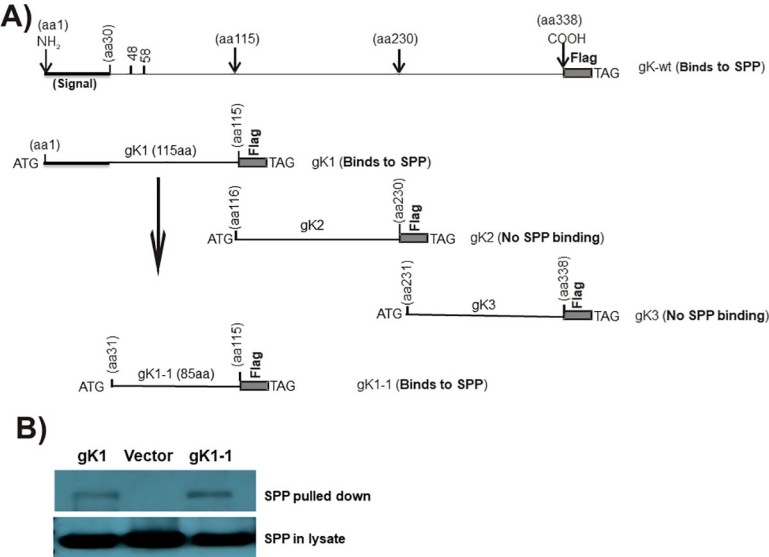

**Fig 1. Mapping of gK binding to SPP.** A) Schematic diagram of gK constructs used for binding to SPP. Full-length gK
is shown at the top. gK1, gK2, and gK3 represent the 1st, 2nd, and 3rd regions of full-length gK, respectively. gK1.1 is
similar to the gK1 fragment except that it is lacking its 30 aa signal sequence. All constructs have an ATG and a
termination codon (TAA) and are inserted into pcDNA3.1 with 3X in-frame Flag tags. B) Binding of gK to SPP *in
vitro*. HeLa cells were transfected with Flag-gK1 or Flag-gK1.1 and HA-SPP plasmids at a 1:1 ratio for 48 hr. Cell
lysates were incubated with anti-Flag antibody bound to IgG beads and the resulting IP was analyzed by western blot
using anti-HA antibody. Lower blot shows similar SPP expression in all three samples using anti-HA antibody for the
western blot.

Flag-gK1.1 and HA-SPP (Fig 1B, gK1.1, upper blot). Thus, the region of gK that binds to SPP
lies between aa31 and aa115 of full-length gK. Cells transfected similarly with vector alone
showed no band (Fig 1B, Vector, upper blot) and levels of SPP in the control blot were similar
between the three groups (Fig 1B, lower blot).

To further fine map the region of gK that binds to SPP, we synthesized a panel of 33 over-
lapping HSV-1 gK peptides (15-mers with 5aa overlaps) (Table 1) as we described previously
[20]. HeLa cells co-transfected with Flag-gK1.1 and HA-SPP were harvested 48 hr after trans-
fection. Cell lysates were mixed with each peptide and IP and western blots were performed as
in Fig 1B. Incubation of co-transfected SPP and gK1.1 with peptides 4, 5, and 6 showed that
peptide 4 significantly reduced gK binding to SPP (Fig 2A, upper blot). The lower control blot
shows similar level of gK was pulled down by anti-flag antibody in each sample (Fig 2A, lower
blot). Only peptide 4, and none of the remaining 32 peptides, blocked gK binding to SPP using
either full-length gK or the gK1.1 fragment. The above results suggest that the secondary struc-
ture of gK does not contribute to its efficient binding to SPP as we previously reported [20,21].

This result suggests that gK4 binds to SPP and comparison of the peptide 3, 4, and 5 amino
acid sequences identified the unique **RCIYA** aa sequence within peptide 4 (Table 1). Based on
the gK4 aa sequence, we synthesized a plasmid with an in-frame Flag sequence (Flag-gK4) as
well as a construct in which the RCIYA aa sequence was mutated to AAAAA with an in-frame
Flag sequence (Flag-gK4 mutant) as described in the Materials and Methods. HeLa cells were
co-transfected with Flag-gK4 and HA-SPP or Flag-gK4 mutant and HA-SPP. IP and western
blots were performed on transfected cell lysates as in Fig 1B. As expected, SPP was immuno-
precipitated by anti-HA-SPP antibody in cells transfected with Flag-gK4 and HA-SPP (Fig 2B,
Flag-gK4, upper blot). In contrast, SPP was not immunoprecipitated by anti-HA-SPP antibody
in cells transfected with Flag-gK4 mutant and HA-SPP (Fig 2B, Flag-gK4 mutant, upper blot).

**Table 1. Fine mapping of the gK binding region to SPP using gK synthetic peptides[a].**

| # | Peptide | aa region | Binds to SPP |
|---|---------|-----------|--------------|
| 1 | MLAVRSLQHLSTVVL | 1–15 | No |
| 2 | STVVLITAYGLVLVW | 10–25 | No |
| 3 | LVLVWYTVFGASPLH | 20–35 | No |
| 4 | ASPLH**RCIYA**VRPTG | 30–45 | Yes |
| 5 | VRPTGTNNDTALVWM | 40–55 | No |
| 6 | ALVWMKMNQTLLFLG | 50–65 | No |
| 7 | LLFLGAPTHPPNGGW | 60–75 | No |
| 8 | PNGGWRNHAHICYAN | 70–85 | No |
| 9 | ICYANLIAGRVVPFQ | 80–95 | No |
| 10 | VVPFQVPPDAMNRRI | 90–105 | No |
| 11 | MNRRIMNVHEAVNCL | 100–115 | No |
| 12 | AVNCLETLWYTRVR | 110–125 | No |
| 13 | TRVRLVVVGWFLYLA | 120–135 | No |
| 14 | FLYLAFVALHQRRCM | 130–145 | No |
| 15 | QRRCMFGVVSPAHKM | 140–155 | No |
| 16 | PAHKMVAPATYLLNY | 150–165 | No |
| 17 | YLLNYAGRIVSSVFL | 160–175 | No |
| 18 | SSVFLQYPYTKITRL | 170–185 | No |
| 19 | KITRLLCELSVQRQN | 180–195 | No |
| 20 | VQRQNLVQLFETDPV | 190–205 | No |
| 21 | ETDPVTFLYHRPAIG | 200–215 | No |
| 22 | RPAIGVIVGCELMLR | 210–225 | No |
| 23 | ELMLRFVAVGLIVGT | 220–235 | No |
| 24 | LIVGTAFISRGACAI | 230–245 | No |
| 25 | GACAITYPLFLTITT | 240–255 | No |
| 26 | LTITTWCFVSTIGLT | 250–265 | No |
| 27 | TIGLTELYCILRRGP | 260–275 | No |
| 28 | LRRGPAPKNADKAAA | 270–285 | No |
| 29 | DKAAAPGRSKGLSGV | 280–295 | No |
| 30 | GLSGVCGRCCSIILS | 290–305 | No |
| 31 | SIILSGIAVRLCYIA | 300–315 | No |
| 32 | LCYIAVVAGVVLVAL | 310–325 | No |
| 33 | VLVALHYEQEIQRRL | 320–335 | No |

[a]A panel of 33 overlapping HSV-1 gK peptides (15-mers with five-amino-acid overlaps) based on the full-length 338 amino acids of gK was used to fine map the region of gK that binds to SPP as we described previously [9,14].

However, SPP levels were similar in cell lysates transfected with Flag-gK4 and Flag-gK4 mutant plasmids (Fig 2B, lower blot). These results suggest that the region of gK that binds to SPP is located within the RCIYA aa region of gK.

## gK peptide 4 reduces HSV-1 replication *in vitro*

As described above, fine mapping results suggest that gK peptide 4 can block binding of gK to SPP (Figs 1 and 2 and Table 1). We previously showed that SPP dominant-negative and SPP inhibitors reduced HSV-1 infectivity *in vitro* [8,9]. In addition, we have shown that depleting SPP in tamoxifen-inducible mice reduced virus infectivity *in vivo* [10]. Supporting these studies, Figs 1 and 2 and Table 1 show that gK4 peptide blocks gK binding to SPP. Thus, we tested

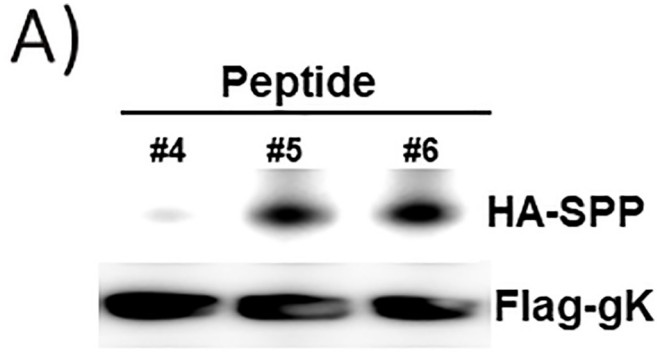

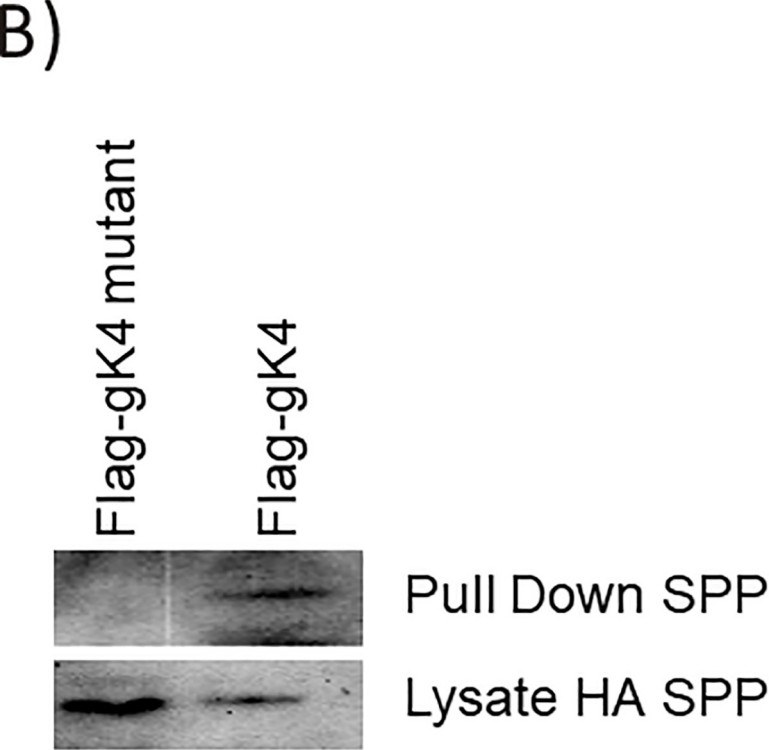

**Fig 2. gK4 peptide inhibits the binding of gK protein to SPP protein.** A) Blocking gK binding to SPP. HEK293 cells were co-transfected with Flag-gK and HA-SPP plasmids. Cells were harvested 48 hr after transfection. Cell lysate was incubated with the indicated peptide for 2 hr at 4˚C. IP using anti-Flag antibody in the presence of 5 μg/ml of indicated peptide recovered similar amounts of Flag-gK protein, but significantly less HA-SPP was recovered in the presence of peptide 4. B) Mutated form of gK4 does not precipitate SPP. HEK293 cells were transfected with HA-SPP and Flag-gK or Flag-gK mutant (amino acid RCIYA was mutated to AAAAA). Cell lysates were immunoprecipitated using anti-Flag antibody. The Flag-gK4 mutant failed to pull down HA-SPP, indicating the importance of the mutated region for gK binding to SPP. Panels: A) Mapping gK binding region to SPP; and B) Fine mapping of gK4 binding to SPP.

if blocking gK binding to SPP using gK#4 results in reduced HSV-1 replication *in vitro*. In order to increase penetration of gK#4 peptide into the cell, peptide #4 was fused to HIV Tat on its N-terminus (**YGRKKRRQRRR**ASPLHRCIYAVRPTG) to increase penetration into the cell. We also synthesized Tat peptide alone (**YGRKKRRQRRR**) for use as a control. To

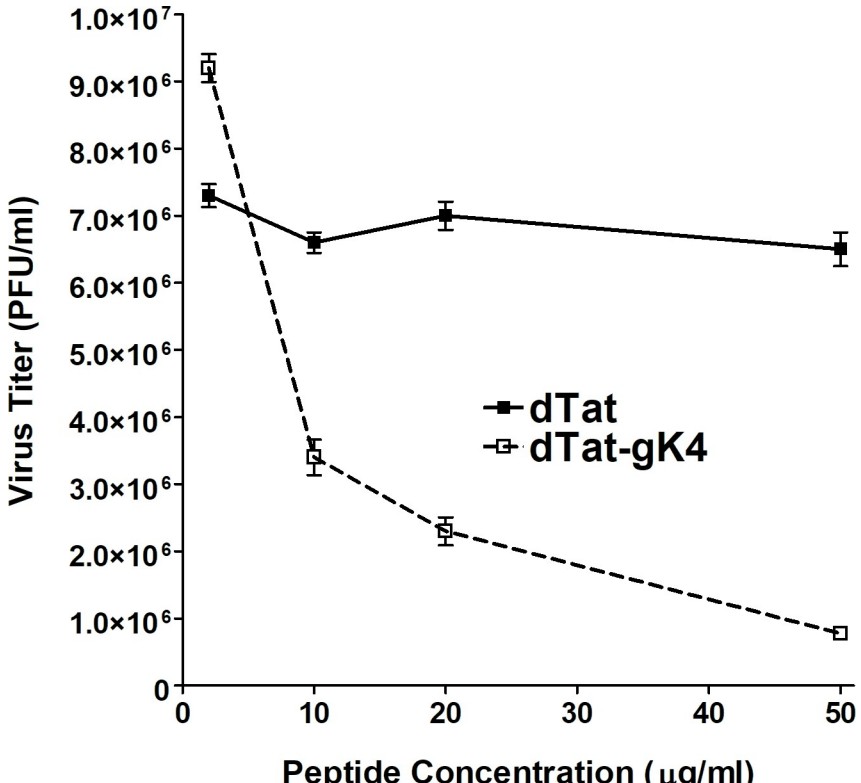

**Fig 3. <u>Inhibition of HSV-1 replication by gK4 peptide *in vitro*</u>.** Vero cells were infected with 0.1 PFU/cell of HSV-1 strain McKrae in the presence of 10, 20, 30, 40, or 50 μg/ml of dTat-gK4 peptide or control dTat peptide for 1 hr. After 1hr infection, the media was replaced with fresh media containing each peptide. Infected cells were harvested 24 hr PI and virus titer was measured by standard plaque assay. Virus replication was significantly inhibited by dTat-gK4 peptide in a dose dependent manner. Half maximal effective concentration (EC 50) of dTat-gK4 peptide was around 2.1 μM (6.6 μg/ml).

increase their stability, these two peptides were synthesized in the amino acid "D" form, and are referred to as dTat-gK4 and dTat. Vero cells were grown to 70% confluency, and 2 hr before infection with HSV-1 strain McKrae, cell media was replaced with DMEM+5% FBS containing 0, 10, 20, or 50 μg/ml of dTat-gK4 or dTat peptide alone (Fig 3). Virus replication in the presence of dTat-gK4 was reduced in a dose dependent manner and was significantly less than in the presence of control dTat peptide (Fig 3, P<0.0001 at all time-points), suggesting that blocking gK binding to SPP reduces virus replication *in vitro*. In contrast to no side effect associated with cell viability at 50 μg/ml concentration of dTat-gK4 peptide, dTat-gK4 peptide at concentration of 100 μg/ml had toxicity on treated RS, 293 and Vero cells.

To confirm that gK4 reduces virus replication *in vitro*, Vero cell monolayers were infected with 0.1 PFU/cell of HSV-GFP virus in the presence 2, 5, 10, or 20 μg/ml of dTat-gK4 or dTat control for 24 hr as described in Materials and Methods. The kinetics of virus replication in the presence of various concentrations of dTat-gK4 or control dTat peptide were quantified by fluorescent-activated cell sorting (FACS) of GFP⁺ cells in three separate experiments (Fig 4A). The percentage of HSV-GFP⁺ cells was reduced significantly in a dose-dependent manner from approximately 40% in cells incubated with 2 μg/ml of dTat-gK4 peptide to 0.6% in the presence of 20 μg/ml of dTat-gK4 peptide (Fig 4A, dTat-gK4). In contrast, incubation of infected cells with different concentrations of dTat control peptide did not significantly reduce HSV-GFP⁺ cells (Fig 4A, dTat). The mean ± SEM of HSV-GFP⁺ cells from three separate

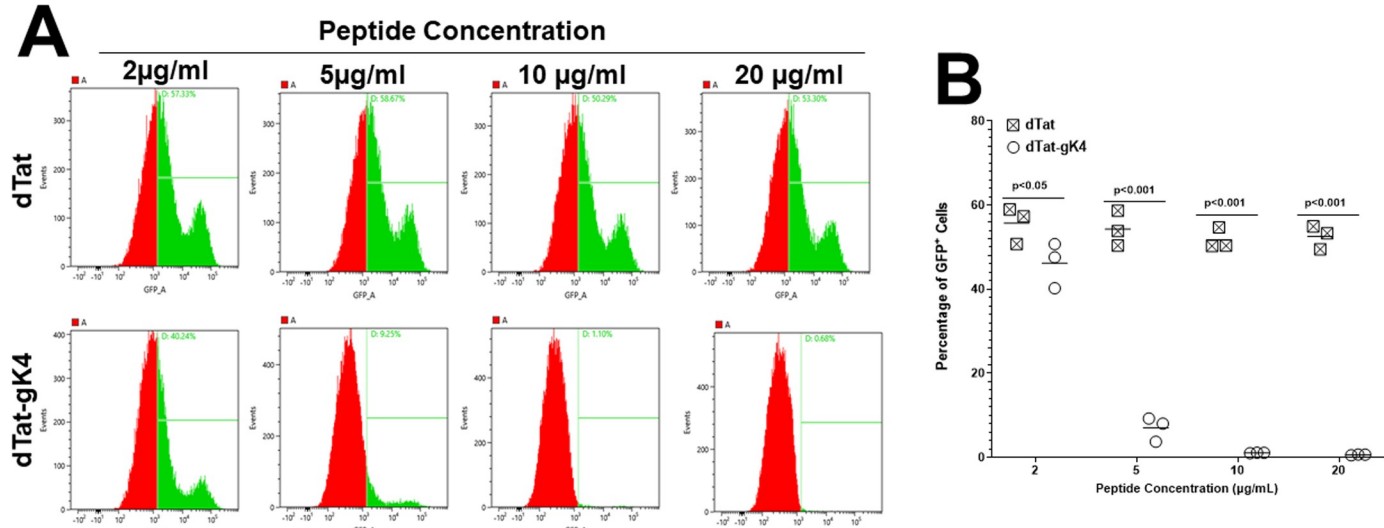

**Fig 4. Analysis of infected cells in the presence of gK4 peptide.** (A) Detection of HSV-GFP<sup>±</sup> cells. Vero cells were infected with 0.1 PFU/cell of HSV-1 expressing GFP in the presence of 2, 5, 10, and 20 μg/ml of dTat-gK4 peptide or control dTat peptide for 1 hr. After 1hr infection, the media was replaced with fresh media containing each peptide. At 24 hr PI, cells were trypsinized, fixed, and the presence of GFP⁺ cells at each peptide concentration was determined by FACS. B) Quantitation of HSV-GFP⁺ cells. Percentage of GFP⁺ cells infected as in (A) above were quantitated by FACS. Each point represents the mean ± SEM from four independent experiments.

FACS analyses are shown in Fig 4B. The number of HSV-GFP⁺ cells was significantly less after incubation with dTat-gK4 than after incubation with the dTat control at all peptide concentrations (Fig 4B, p<0.001).

## dTat-gK4 Peptide alters subcellular localization of gK

Because dTat-gK4 peptide significantly reduced HSV-1 replication *in vitro* using two different approaches and two different HSV-1 strains, we asked whether dTat-gK4 peptide alters gK localization. We infected Vero cells with 1 PFU/cell of HSV-1 strain VC1 in the presence of 20 μg/ml of dTat-gK4 or dTat control peptide. At 16 hr post infection (PI), cells were fixed, permeabilized, and immunostained with anti-V5 (to detect gK) and anti-GM130 (to detect the Golgi complex) (Fig 5A). In cells treated with dTat control peptide, gK protein was strongly expressed and more cells were positive than in cells incubated with dTat-gK4, which also had a weaker signal and fewer cells were positive (Fig 5A, gK). GM130 expression was not affected by dTat-gK4 peptide and was similar to that in dTat treated cells (Fig 5A, GM130). As expected, in the presence of dTat control peptide, gK colocalized with GM130 protein, while in the presence of dTat-gK peptide, gK protein did not colocalize with GM130 (Fig 5A, Merge). The number of gK⁺ cells in the presence of dTat-gK4 and dTat control peptides from the above infected cells was counted per multiple visual fields. Approximately 75% of infected cells in the presence of dTat were gK⁺ compared with 22% in the presence of dTat-gK4 (Fig 5B). Thus, dTat-gK4 treated Vero cells had significantly less gK⁺ cells than did dTat treated control cells (Fig 5B, p<0.01, Fisher exact test). Thus, blocking gK binding to SPP alters gK expression and localization.

## dTat-gK4 Peptide reduces virus replication in the eye of ocularly infected BALB/c mice

The *in vitro* results described above suggest that blocking gK binding to SPP using dTat-gK peptide reduces virus replication in infected Vero cells (Figs 4–6). Thus, we sought to

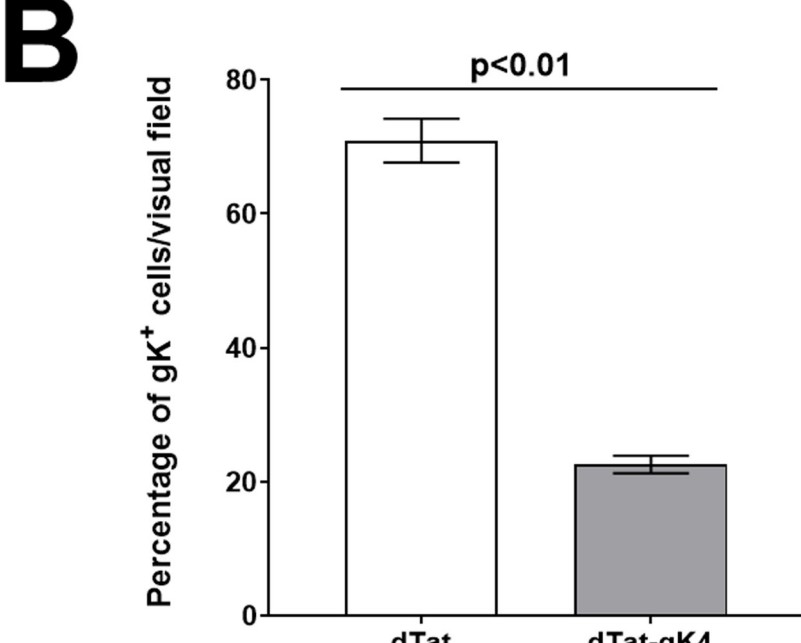

**Fig 5. Effect of blockade of gK interaction with SPP using gK4 on gK localization in HSV-1 infected cells.** A) Detection of HSV-1 gK in infected cells. Vero cells grown to confluency on chamber slides were infected with 1 PFU/cell of VC1 virus in the presence of 20 μg/ml of dTatgK4 or dTat peptide for 1 hr. After 1hr infection, the media was replaced with fresh media containing each peptide for 16 hr. Infected cells were fixed and immunostained using anti-V5 (for gK) with anti-GM130 (for Golgi), and chicken anti-goat AlexaFluor 488 (for gK) with chicken anti-rabbit AlexaFluor 594 (for GM130) as secondary antibodies. DAPI was used for nuclear staining (blue). There were fewer gK positive cells in the dTat-gK4 treated group than in control dTat peptide-treated group. gK protein colocalized with Golgi protein GM130 in the control peptide group, indicating enrichment of gK protein in the Golgi apparatus. However, gK protein did not colocalize with GM130

in the dTat-gK4 treatment group. gK protein localization to the Golgi was inhibited by blocking the binding of gK to SPP. Photomicrographs are shown at 630X direct magnification. Experiments were repeated twice; and B) Quantification of photomicrographs from A. Different areas of 3 slides/peptide from IHC described above were imaged and the number of HSV-1 gK⁺ cells was counted. Each point represents the mean ± SEM of HSV-1 gK⁺ DCs from 7 images.

determine if this peptide also reduces HSV-1 replication *in vivo*. dTat-gK4 or dTat peptide were given to BALB/c mice as an eye drop in two separate experiments as described in Materials and Methods. Following ocular infection with $1 \times 10^5$ pfu/eye of HSV-1 strain McKrae, virus replication significantly decreased in the eyes of dTat-gK4 treated mice on days 2–4 PI (Fig 6A; P <0.001, Student t-test compared to dTat control). Thus, blocking the binding of viral gK to cellular SPP using dTat-gK4 as an eye drop significantly reduced virus replication in the eye of infected mice suggesting its possible therapeutic potential.

Herpetic blepharitis is an inflammation of the lid margin following ocular HSV-1 infection, and in mice, increased blepharitis usually correlates with increased HSV-1 replication in the eye. Blepharitis was measured 4 days after ocular infection as described in Materials and Methods. dTat-gK4 treated BALB/c mice had significantly less blepharitis than did dTat treated control mice (Fig 6B, p = 0.028, Mann Whitney test). These results suggest that reduced ocular virus replication after treatment with dTat-gK4 (Fig 6B) correlated with lower blepharitis.

After treatment with dTat-gK4 or dTat control, groups of ten BALB/c mice from two separate experiments were infected ocularly with $1 \times 10^5$ PFU/eye of HSV-1 strain McKrae. Survival in dTat-gK4 mice was significantly higher than in mice treated with dTat peptide (5 of 15 versus 1 of 15) (Fig 6C; p = 0.027, Fisher's exact test). Thus, blocking gK binding to SPP using dTat-gK4 peptide appeared to provide more protection of naive mice against death than did dTat peptide in the treated control group. This increased survival in dTat-gK4 treated mice directly correlated with lower virus replication in the eye of treated-infected mice.

## Expression of HSV-1 transcripts gK, gB, ICP0, and UL20 are reduced during primary ocular infection of mice in the presence of dTat-gK4 peptide

The above results suggest that ocular virus replication and blepharitis were significantly reduced in the presence of dTat-gK4 peptide, while survival increased (Fig 6). To determine if blocking gK binding to SPP also affects the levels of different viral transcripts *in vivo*, BALB/c mice were treated with dTat-gK4 peptide or dTat control peptide as above and infected with $1 \times 10^5$ PFU/ eye of HSV-1 strain McKrae. Corneas and TG were collected on day 4 PI, and total RNA was isolated and subjected to TaqMan qRT-PCR to determine the copy numbers for gK, gB, ICP0, and UL20 mRNAs. GAPDH mRNA in each sample was used as an internal control. As we saw with virus replication in the eye, we found significantly lower levels of gK (Fig 7A, p<0.05), gB (Fig 7B, p<0.05), ICP0 (Fig 7C, p<0.05), and UL20, (Fig 7D, p<0.05) transcripts in the corneas of mice that received dTat-gK4 peptide than control mice that received dTat peptide.

Similar to the above results for cornea, expression level of gK (Fig 7E, p<0.05), gB (Fig 7F, p<0.05) and ICP0 (Fig 7G, p<0.05) transcripts were significantly lower in the TGs of mice that received dTat-gK4 peptide than control mice that received dTat peptide.

Collectively, these results show that application of dTat-gK4 peptide as an eye drop significantly reduced viral transcription in corneas and TG of ocularly infected mice.

## Blocking gK binding to SPP by dTat-gK4 peptide did not alter cellular gene expression in corneas or TG of infected mice

The above results suggest that blocking gK binding to SPP by dTat-gK4 peptide reduced virus replication and viral transcripts in corneas and TG of treated mice (Figs 6 and 7). To

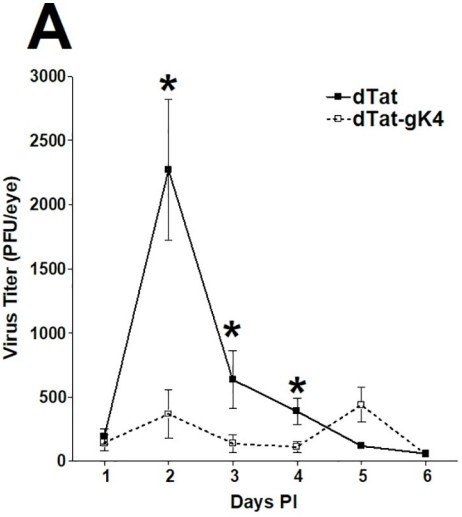

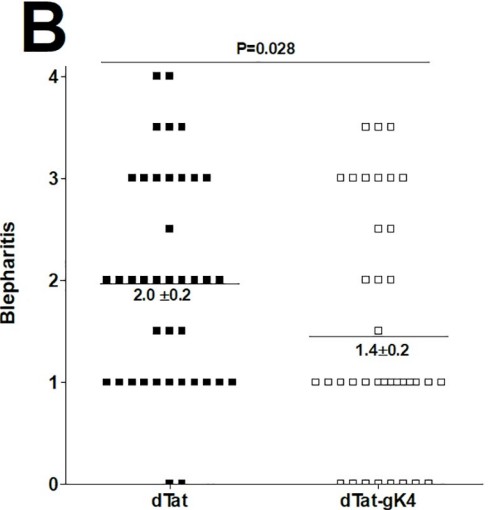

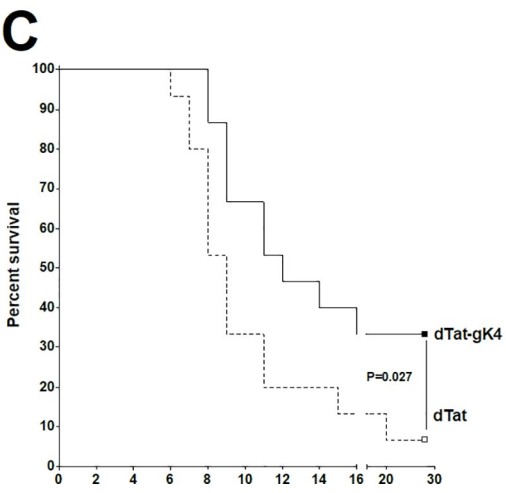

**Fig 6. Effect of dTat-gK4 peptide on virus replication, eye disease, and survival in BALB/c infected mice.** (A) Virus replication in the eye of infected mice. Ten BALB/c mice from two separate experiments were infected ocularly with 1 X $10^5$ PFU/eye of HSV-1 strain McKrae. One day before infection and on days 1–5 PI, mice received 20 μg of dTat-gK4 or dTat control peptide as an eye drop twice daily in 5 μl of 1xPBS. Eye swabs were collected each day for 6 days and virus titer in tear films were measured by standard plaque assay. Each point represents mean ± SEM from 20 eyes; (B) Blepharitis in ocularly infected mice. Blepharitis in infected mice was measured on day 4 PI. Each point is mean ± SEM from 20 eyes; and (C) Survival in ocularly infected mice. Survival of the above mice was followed for 28 days. Survival is based on ten mice per treatment group.

determine whether treating mice with dTat-gK4 peptide affected cellular transcript levels in corneas and TG of infected mice, BALB/c mice, after treatment with dTat-gK4 peptide or dTat control peptide, were infected ocularly as above with HSV-1 strain McKrae. Corneas and TG were collected on day 4 PI, total RNA extract was isolated, and relative levels of CD4, CD8, IFNα2, IFNβ, and IFNγ transcripts were determined using TaqMan qRT-PCR. The results are presented as "fold" increase (or decrease) compared to baseline mRNA levels in corneas and TG of uninfected naive mice. GAPDH mRNA in each sample was used as an internal control. Levels of CD4 and CD8 (Fig 8A) as well as IFNα2, IFNβ, and IFNγ (Fig 8B) mRNA expression were similar in corneas of mice treated with dTat-gK4 and control dTat peptide (p>0.05).

Levels of CD4 and CD8 (Fig 8C) as well as IFNα2, IFNβ, and IFNγ (Fig 8D) gene expression levels were lower in TG of mice treated with dTat-gK4 than with control dTat peptide but these differences were not statistically significant (p>0.05). Overall, levels of CD4, CD8, IFNα2, IFNβ, and IFNγ were higher in TG of infected mice than in corneas (Fig 8, compare A and B with C and D). Further, CD8 transcript levels were higher than CD4 transcripts in both corneas and TG of infected mice (Fig 8, compare A with C). Thus, treating mice with dTat-gK4 peptide significantly reduced virus replication in both eyes and TG of infected mice without having any negative effects on the level of the tested cellular transcripts in treated mice.

## Effect of dTat-gK4 peptide on virus replication, survival, and latency in C57BL/6 mice

Our *in vivo* results with BALB/c mice suggest that blocking gK binding to SPP using dTat-gK4 peptide reduces virus replication, eye disease, and death in infected mice (Figs 6 and 7). Because BALB/c mice are highly susceptible to ocular infection with virulent HSV-1 strain McKrae, we next asked if, similar to BALB/c mice, dTat-gK4 peptide also affects HSV-1 infectivity in C57BL/6 mice that are refractory to HSV-1 strain McKrae even at higher infection doses. Thus, we treated 13 C57BL/6 mice with dTat-gK4 peptide and 12 with dTat peptide as an eye drop in two separate experiments as described in Materials and Methods. Treated mice were ocularly infected with HSV-1 strain McKrae (2 X $10^5$ PFU/eye). Tear films were collected daily from both eyes of ten mice on days 1 to 5 PI, and the amount of virus in each eye was determined by standard plaque assays. All mice that received dTat-gK4 peptide had lower peak virus titers/eye than did dTat control treated mice on days 2, 3 and 4 PI (Fig 9A; p <0.05). Taken together, and similar to our BALB/c mice results described above (Fig 7), these findings with C57BL/6 mice suggest that administration of dTat-gK4 peptide very effectively promotes clearance of HSV-1 from the eyes of infected mice. Survival of the treated mice described above was followed for 4 weeks with a similar survival pattern in both experiments. All 13 mice in dTat-gK4 group survived ocular challenge, while 10 of 12 dTat peptide-treated mice survived ocular infection (p>0.05). Thus, similar to the results described above with BALB/c mice, reduction of virus replication by dTat-gK4 peptide appears to improve protective responses of mice against lethal HSV-1 infection.

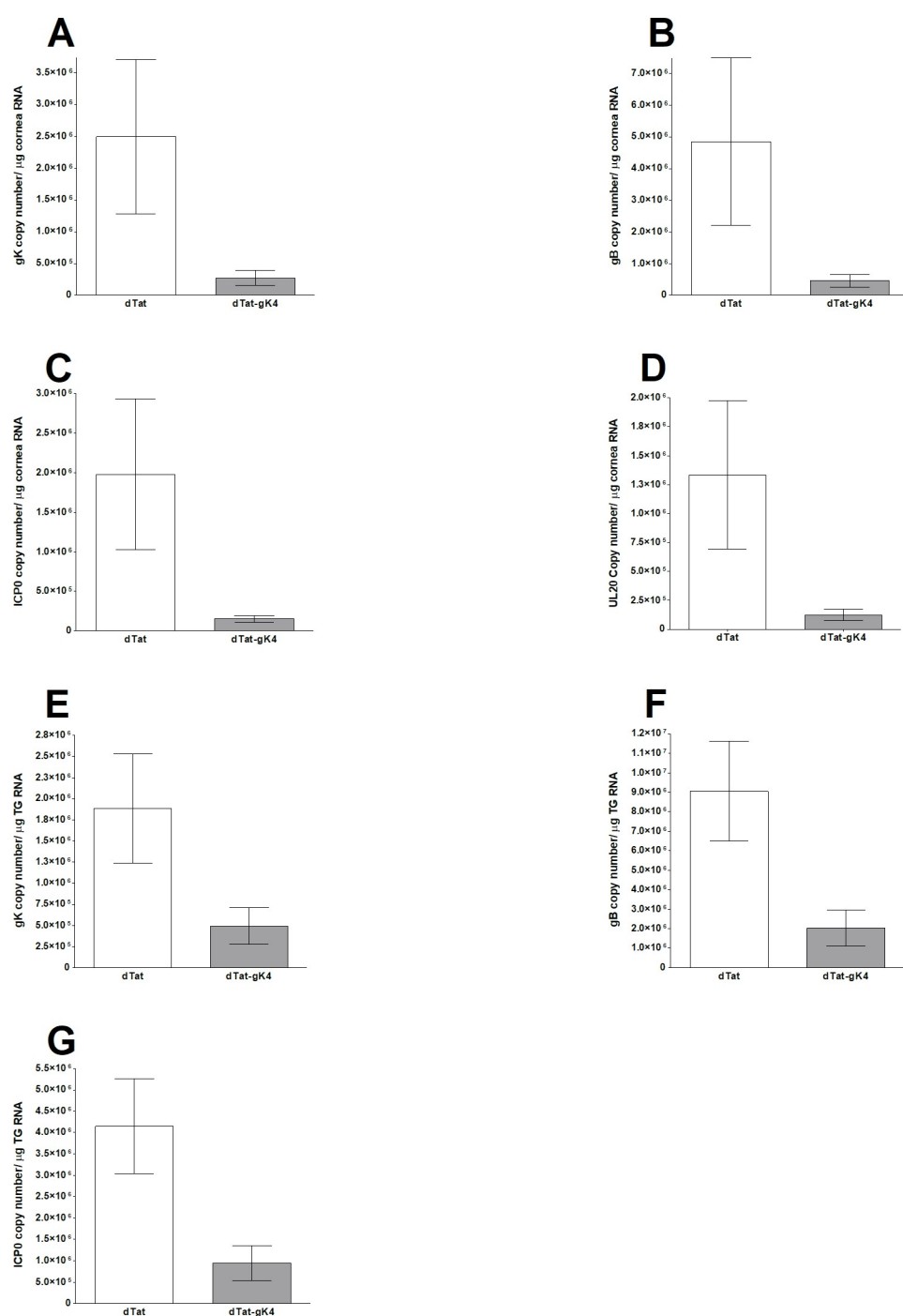

**Fig 7. Expression of gK, UL20, ICP0, and gB transcripts in cornea and TG of ocularly infected mice in the presence of dTat-gK4.** BALB/c mice were ocularly infected with 1 X10$^5$ PFU/eye of HSV-1 strain McKrae and treated with dTat-gK4 or dTat control peptide as described in Materials and Methods. Levels of gK, UL20, ICP0, and gB transcripts in the cornea and TG were determined on day 4 PI by qRT-PCR. Estimated relative copy number of HSV-1 UL20, gK, ICP0 and gB was calculated using standard curves generated from pUL20 [14], pAC-gB1 [65], pAc-gK1 [6], and pcDNA3.1-ICP0 [61]. Briefly, a DNA template, serially diluted 10-fold such that 1 μl contained from 10$^3$ to 10$^{11}$ copies of each plasmid, was amplified by TaqMan PCR with the same primer set. Copy number of each reaction was determined by comparing the $C_T$ of each sample to the threshold cycle of the standard. GAPDH expression was used to normalize relative expression of each transcript in the cornea and TG of infected mice. Each bar represents mean ± SEM from five mouse corneas or TG. Panels: (A) gK expression in cornea; (B) gB expression in cornea; (C) ICP0 expression in cornea; (D) UL20 expression in cornea; (E) gK expression in TG; (F) gB expression in TG; and (G) ICP0 expression in TG.

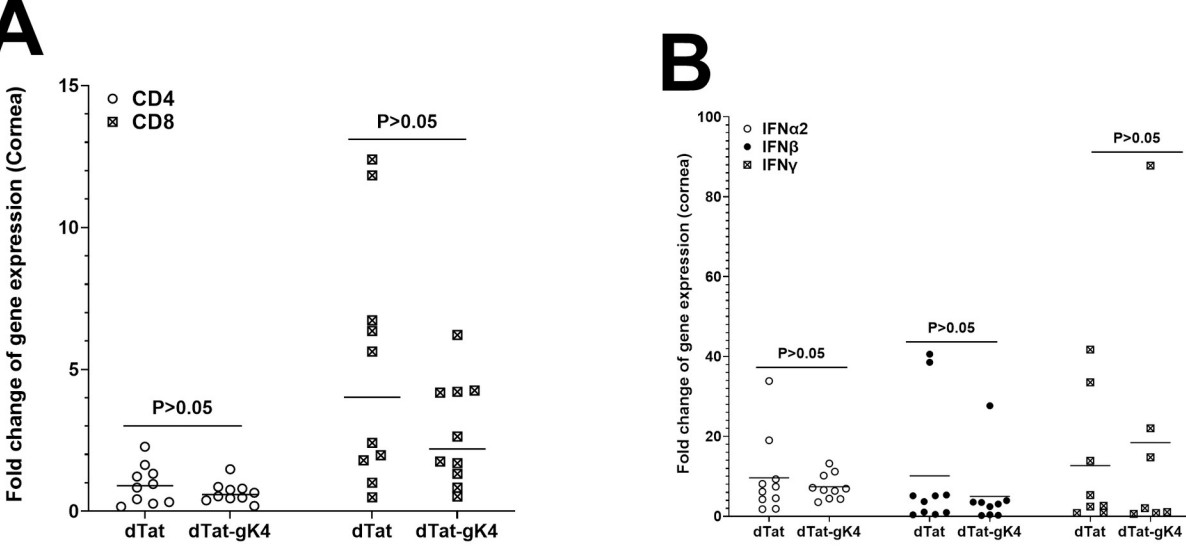

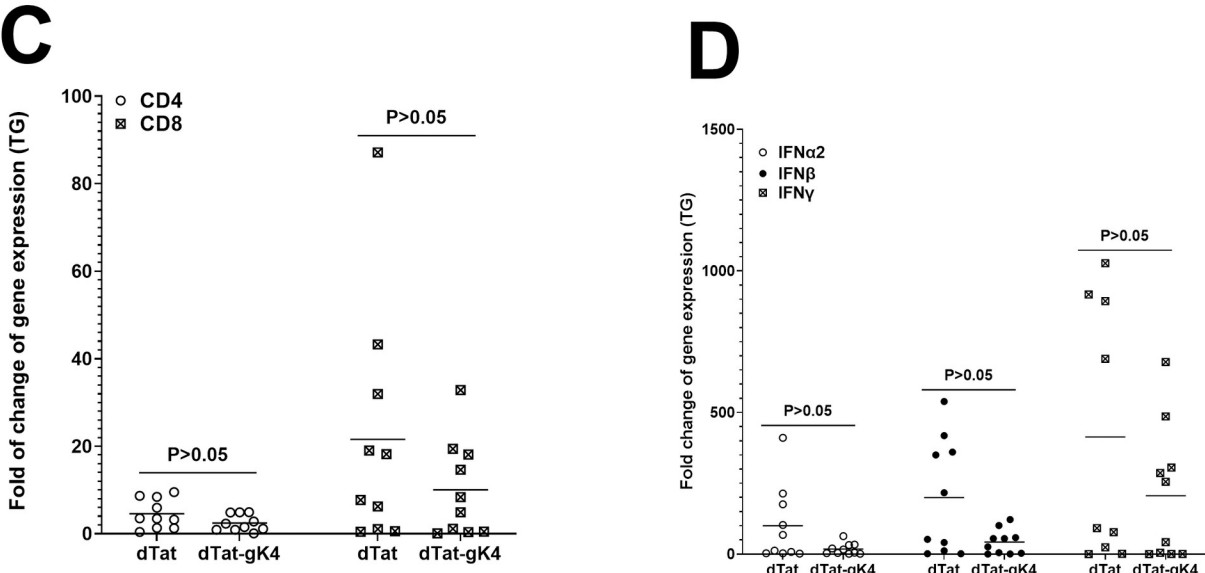

**Fig 8. Effect of dTat-gK4 treatment on expression of CD4, CD8, IFNα2, IFNβ, and IFNγ transcripts in cornea and TG of ocularly infected mice.** BALB/c mice were infected as described in Fig 7. Levels of CD4, CD8, IFNα2, IFNβ, and IFNγ transcripts in cornea and TG were determined on day 4 PI by qRT-PCR. The ratio of expression of each mRNA in the cornea and TG was normalized to its expression in the cornea and TG of uninfected control mice. GAPDH expression was used to normalize relative expression of each transcript in the cornea and TG of infected mice. Each bar represents the mean ± SEM from five mouse corneas or TG. Panels: (A) gK expression in cornea; (B) gB expression in cornea; (C) ICP0 expression in cornea; (D) UL20 expression in cornea; (E) gK expression in TG; (F) gB expression in TG; and (G) ICP0 expression in TG.

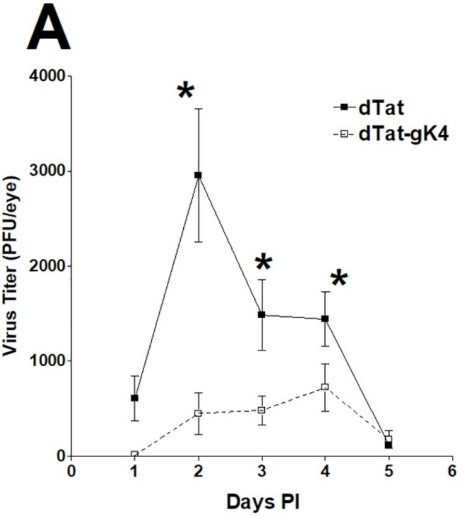

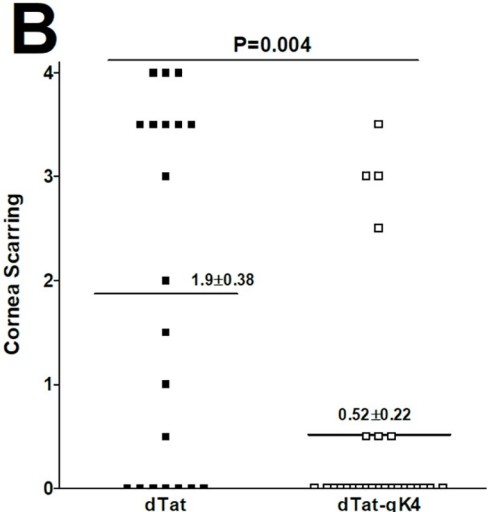

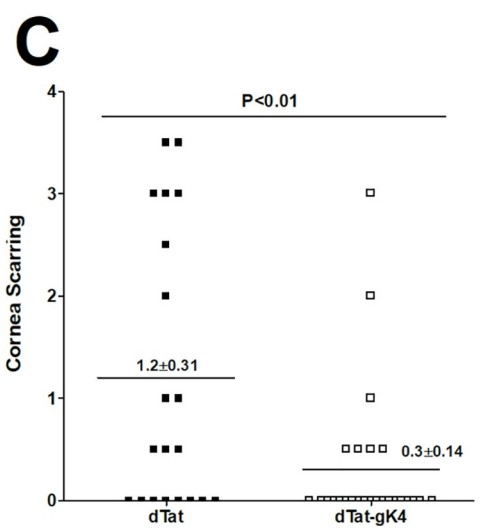

**Fig 9. Effect of dTat-gK4 peptide on virus replication and CS in C57BL/6 infected mice.** C57BL/6 mice from two separate experiments were infected ocularly with 2 X 10⁵ PFU/eye of HSV-1 strain McKrae. One day before infection and on days 1–5 PI, mice received 20 μg of dTat-gK4 or dTat control peptide twice daily as an eye drop in 5 μl of 1xPBS. (A) Virus replication in the eye of infected mice. Eye swabs were collected each day for 6 days and virus titer in the tear film was measured by standard plaque assay. Each point is mean ± SEM from 26 eyes for dTat-gK4 and 22 eyes for dTat control. (B) CS in ocularly infected mice. CS in treated mice was measured on days 14 and 28 PI. Each point is mean ± SEM from 26 eyes for dTat-gK4 and 20 eyes for dTat control. Panels: A) Virus titer in the eye; B) CS on day 14 PI; and C) CS on day 28 PI.

## dTat-gK treatment reduces corneal scarring in ocularly infected mice

To determine the effect of dTat-gK4 treatment on eye disease, the severity of CS on days 14 and 28 PI was evaluated in surviving C57BL/6 mice described above (Fig 9A). On days 14 (Fig 9B, p = 0.004) and 28 (Fig 9C, p<0.01) PI, C57BL/6 mice treated with dTat-gK4 developed significantly less CS than mice treated with dTat control peptide. Thus, treating infected mice with dTat-gK4 peptide significantly reduced virus-induced CS.

## Reduced latency in TG of latently-infected mice treated with dTat-gK4 peptide

The surviving ocularly infected C57BL/6 mice described above (Fig 9) were euthanized on day 28 PI and their TGs were isolated. Total RNA was isolated from these TGs and TaqMan qRT-PCR was performed to quantify latency-associated transcript (LAT) mRNA copy number as described in Materials and Methods. Cellular GAPDH mRNA was used as an internal control. The amount of LAT mRNA during latency in mice treated with dTat-gK4 peptide was significantly lower than in control mice treated with dTat peptide (Fig 10A; p = 0.03, Student's t-test). These results suggest that blocking the binding of gK to SPP with dTat-gK4 peptide reduced latency in the TG of HSV-1 infected mice, which correlates with lower virus replication and lower CS in infected mice.

## dTat-gK4 treatment reduces exhaustion markers in TG of latently-infected mice

We previously showed that increased HSV-1 latency correlates with increased T cell exhaustion markers [22] and, as seen in Fig 10A, dTat-gK4 treatment reduced LAT expression in infected mice. Thus, we investigated the effect of reduced latency on expression of T cell exhaustion markers and their related cytokines in TG of mice latently-infected with HSV-1 after treatment with dTat-gK4 peptide or dTat control peptide. C57BL/6 mice were infected and relative levels of mRNAs of exhaustion markers (PD-1, Tim-3) and cytokines whose levels may be altered by exhaustion (IFNγ, TNF-α), were determined by TaqMan qRT-PCR of total TG extracts. The results are presented as "fold" increase (or decrease) compared to baseline mRNA levels in TG from uninfected naive mice (Fig 10B). In TG of mice treated with dTat-gK4 peptide, levels of PD-1 (Fig 10B, p<0.01), Tim-3 (Fig 10B, p<0.01), IFNγ (Fig 10B, p<0.01), and TNF-α (Fig 10B, p = 0.02) mRNAs were all significantly less than levels in control mice treated with dTat peptide. These results suggest that there are fewer T cell exhaustion markers in TG from latently-infected mice treated with dTat-gK4 peptide than in the dTat peptide-treated control group. Thus, blocking the interaction of gK with SPP using a specific 15mer gK peptide reduced exhaustion markers in treated mice, which correlates with reduced latency and reduced ocular virus replication.

## Discussion

HSV-1 encodes at least 80 genes [15], 12 of which encode glycoproteins [3,4,6,15,23,24]. Previously we expressed ten of these known HSV-1 glycoproteins in a baculovirus system and tested

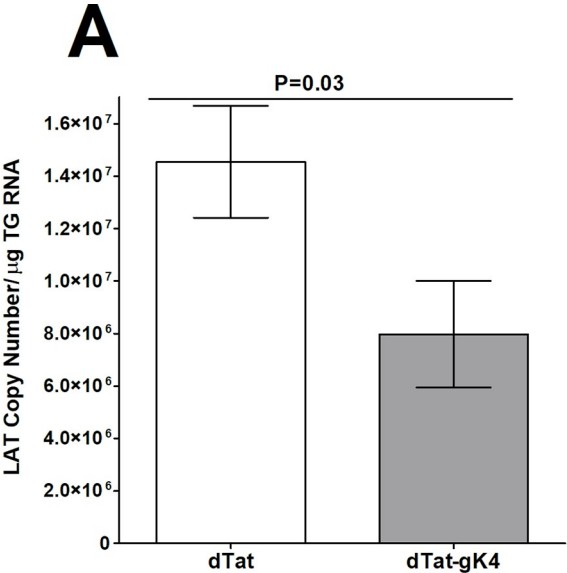

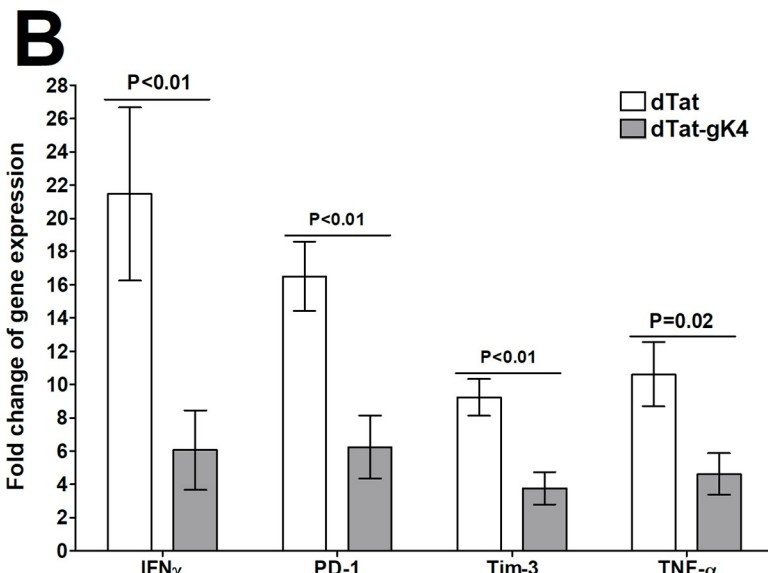

**Fig 10. Effect of dTat-gK4 peptide on latency and exhaustion marker levels in C57BL/6 infected mice.** TG from mice described in Fig 9 legend were harvested on day 28 PI. Total RNA from each TG was isolated as described in Materials and Methods. A) Quantitation of LAT RNA in TG of immunized mice. Quantitative RT-PCR was performed on each mouse TG. In each experiment, an estimated relative copy number of HSV-1 LAT was calculated using standard curves generated from pGem5317. Briefly, DNA template was serially diluted 10-fold such that 1 μl contained $10^3$ to $10^{11}$ copies of LAT, then subjected to TaqMan PCR with the same primer set. By comparing the normalized threshold cycle ($C_T$) of each sample to the threshold cycle of the standard, copy number for each reaction was determined. GAPDH expression was used to normalize relative expression of viral LAT RNA in TG. Each bar represents the mean ± SEM from 22 TG for dTat-gK4 treated mice and 18 TG for dTat control group. B) qRT-PCR analyses of IFNγ, PD-1, Tim-3, and TNF-α transcripts in TG of latently-infected mice. Total RNA described above from each individual TG was used to estimate relative expression of IFNγ, PD-1, Tim-3, and TNF-α transcripts in TG of mice in dTat-gK4 treated or dTat control group. The expression ratio of each mRNA in the TG was normalized to its expression in TG of uninfected control mice. GAPDH expression was used to normalize relative expression of each transcript in TG of immunized mice. Each bar represents the mean ± SEM from 12 TG for each group. Panels: A) LAT; and B) Exhaustion markers.

their immunogenicity in mice [3,4,6,24,25]. Among all of the tested glycoproteins, only gK exacerbated CS while also causing severe dermatitis and proliferation of the eye [6]. HSV-1 can cause eye disease and blindness [26], but which of the more than 80 HSV-1 genes contribute to pathology is not well-known. Our numerous publications over the past 30 years suggest that HSV-1 gK contributes to this pathology independent of virus or mouse strain [3,4,7,20,21,27].

Since gK is associated with HSV-induced pathology, we recently showed that gK binds to SPP and that SPP is required for HSV-1 infectivity *in vitro* and *in vivo* [9,10]. We have also shown that inhibitors of SPP reduced virus infectivity and eye disease *in vivo* [8]. SPP (aka minor histocompatibility antigen H13) is an ER-resident protein [28–33], and is highly conserved among different species [34–37]. There is a 96% amino acid homology between human and mouse SPP [38]. SPP knockout mice are embryonic lethal [39]. Thus, to overcome this issue, we generated tamoxifen-inducible Cre recombinase mice and showed that depletion of SPP significantly reduced HSV-1 replication in the eye as well as reduced latency in ocularly infected mice [10].

It is estimated that of the 70–90% of American adults with antibodies to HSV-1 and/or HSV-2, about 25% have clinical symptoms [26,40–42], with HSV-1 being responsible for >90% of ocular HSV infections. In addition, 15–50% of primary genital herpes is caused by HSV-1 and recent studies indicate that the proportion of clinical first episode genital herpes due to HSV-1 is increasing [43–45]. Despite the serious consequences of recurrent ocular herpes infection, no drug to prevent ocular recurrence has been approved by the FDA. Thus, there is a critical need to develop alternative approaches to prevent and control serious HSV-1-induced ocular diseases. Our approach to this issue has been to identify molecular features of the virus that drive the disease process. Our ongoing studies strongly suggest that glycoprotein K (gK) is the viral gene that exacerbates eye disease in mice and rabbits [4,5,21] as well as in humans [27]. We have now pinpointed specific interactions of gK with SPP that appear to play critical roles in viral replication in the cornea and in eye disease [8–10,13,14].

Using two different strains of HSV-1, we found that dTAT-gK4 significantly reduced virus replication *in vitro* in a dose dependent manner using three complementary approaches. The dTat-gK4 peptide also altered both gK expression and its cell surface localization. We also investigated the *in vivo* effects of SPP blockage using gK peptide in ocularly infected mice. Blocking gK binding to SPP significantly reduced virus replication in the eye of both BALB/c and C57BL/6 mice. We also found significantly lower levels of gK, UL20, ICP0, and gB transcripts in the corneas and TG of infected mice treated with dTat-gK4 peptide than in control mice. Similarly, we recently reported that depletion of SPP significantly reduced virus replication in the eye of ocularly infected mice below that of WT mice [10]. Further, we previously reported that *GODZ⁻/⁻* infected mice, lacking GODZ expression, had less virus replication in the eye and less gK, UL20, and gB transcripts in the cornea than did controls [13]. In contrast to the lower levels of viral transcripts in cornea and TG of uninfected mice treated with dTat-gK4 peptide, treatment of infected mice with dTat-gK4 peptide did not alter expression of some cellular genes involved in innate and adaptive protection. In contrast, we detected lower levels of immune infiltrates in corneas of mice lacking SPP [10]. These differences in levels of immune infiltrates in the cornea and TG of dTat-gK4 treated mice were similar to SPP-deficient mice, suggesting that while blockage of gK binding to SPP reduced virus replication in the eye and TG, it did not affect levels of cellular infiltrates in infected mice suggesting that the dTat-gK4 peptide does not have a negative effect on host genes. Thus, the protective effect of dTat-gK4 directly correlates with viral processing by SPP and is independent of the host immune response to treatment. In this study we have used TAT peptide as a control and we did not observe any antiviral activity associated with using three different strains of HSV-1,

three different cell lines and two different strains of mice. Similar to this study, TAT alone did not have any antiviral activity against two different strains of HSV-1, adenovirus type 5, cyto-megalovirus, vaccinia virus, and simian virus 40 in cell culture models [46]. In addition, no antiviral activities associated with the TAT control peptide was reported for influenza virus [47,48], Ebola virus [49], Epstein-Barr virus [50], or Japanese encephalitis virus [51].

BALB/c mice are highly susceptible to infection with virulent HSV-1 strain McKrae and blockage of gK binding to SPP by dTat-gK4 peptide did affect their susceptibility to ocular infection. In contrast to BALB/c mice, C57BL/6 mice are refractory to HSV-1 strain McKrae and in contrast to control treated C57BL/6 mice, dTat-gK4 treated C57BL/6 mice were completely protected from death. We previously showed a direct relationship between input virus, the duration of primary virus replication in the eye, and the severity of eye disease in ocularly infected BALB/c mice [3,4,6]. Treatment of BALB/c mice with dTat-gK4 peptide reduced the severity of blepharitis compared with control mice and the severity of blepharitis directly correlates with levels of virus replication in the eye. Similar to BALB/c mice, levels of CS following infection of C57BL/6 mice treated with dTat-gK4 peptide were significantly lower than in control mice. In this study, all our *in vivo* experiments were done using HSV-1 strain McKrae which does not need corneal scarification for effective eye disease investigations, thus heightening the clinical significance of this study.

Latency was significantly reduced in dTat-gK4 peptide-treated mice compared to controls, suggesting that reduced latency in the absence of gK binding to SPP correlates with reduced virus replication in the eye and TG of treated mice. This study confirms our previous results that showed a direct correlation between reduced virus replication in the eye and reduced latency in infected mice [10,52].

Previously, we reported that TG with more LAT RNA tended to have more PD-1 and Tim-3 mRNAs [22,53]. Here we found that treatment with dTat-gK4 peptide correlated with reduced LAT expression, lower PD-1 and Tim-3 T cell exhaustion markers, and overall reduced TNF-α and IFNγ cytokine production in TG of dTat-gK4 treated mice. Thus, treatment with dTat-gK4 appears to decrease latency, which correlates with reduced T cell exhaustion.

In summary, we have provided the first evidence that blocking gK binding to SPP reduces virus replication and latency, thereby reducing the pathology associated with gK and SPP interaction without affecting the host. This study provides new insights into the determinants of HSV-1 replication and infectivity and provides a framework for the design of drugs that will offer new ways to combat herpesvirus infections with minimal to no side effects. In addition to the use of acyclovir and ganciclovir, nanocarriers, prodrugs, peptide conjugation, and in situ gelling systems have been used to slow the onset or progression of HSK and HSV-1 replication *in vitro* and in animal models [40,41,54–60]. Therefore, these studies and many more have evaluated the antiviral activities of natural or synthesized compounds as potential treatments for viral keratitis. However, to our knowledge this is the first study showing that inhibition of SPP-gK interaction with a synthetic gK peptide has antiviral activity. In this study we examined the prophylactic potential of blocking gK-SPP interaction, but since most adults are seropositive to HSV-1 infection, further studies will be needed to test the therapeutic potential of blocking this interaction. However, using gK4 as an eye drop outside of a clinic setting will be easy to do.

## Material and methods

### Ethics statement

All animal procedures were performed in strict accordance with the Association for Research in Vision and Ophthalmology Statement for the Use of Animals in Ophthalmic and Vision

Research (https://www.arvo.org/About/policies/statement-for-the-use-of-animals-in-ophthalmic-and-vision-research/) and the NIH *Guide for the Care and Use of Laboratory Animals.* The animal research protocol was approved by the Institutional Animal Care and Use Committee of Cedars-Sinai Medical Center (Protocol #9129).

## Cells, virus, and mice

Vero, HeLa, and HEK293 cells were cultured in DMEM plus 10% fetal bovine serum and maintained as described previously [9,13,14,61]. RS cells were cultured in MEM plus 5% fetal bovine serum. Triple-plaque-purified HSV-1 strain McKrae was grown in RS cell monolayers as described previously [4]. The VC1 virus with V5-tagged gK with HSV-1 F background (a gift from Konstantin G Kousoulas; Division of Biotechnology and Molecular Medicine, School of Veterinary Medicine, Louisiana State University, Baton Rouge, Louisiana, USA) was described previously [62]. HSV-GFP+ (a gift from Peter O'Hare; Marie Curie Research Institute, Surrey, United Kingdom) is a recombinant virus with HSV-1 strain 17 background that contains the gene encoding a major tegument protein, VP22, linked to green fluorescent protein (GFP) [63,64]. HSV-GFP+ virus and VC1 viruses were propagated as we described previously [13,14]. Six to eight week old female C57BL/6 and BALB/c mice were purchased from The Jackson Laboratory (Bar Harbor, ME). We used female mice in this study, since in numerous studies that we have done we did not detect any significant differences between male and female mice ocularly infected with HSV-1.

## gK plasmids for fine mapping of the gK binding site to SPP

A schematic diagram of full-length gK with an in-frame Flag tag at the carboxy terminus and four fragments of gK constructs with an ATG, an in-frame Flag tag at the carboxy terminus and a termination codon used in this study are shown in Fig 1. Full-length SPP with an in-frame HA tag and ER retention signal were described previously [9]. All these constructs were synthesized (GenScript, Piscataway, NJ) and inserted into BamHI site of pcDNA3.1 and sequences were verified with standard dideoxy sequencing.

## gK peptide synthesis

A panel of 33 overlapping HSV-1 gK peptides (15-mers with five-amino-acid overlaps) were synthesized by Mimotopes (San Diego, CA) as we described previously [20] (Table 1). These 33 peptides cover the entire gK protein sequence. Purity of original peptides synthesized was at least 85%. All peptides were dissolved in DMSO at a concentration of 1 µg/µl and stored at –20˚C. These peptides were used to fine map the gK binding site to SPP.

## Synthesis of specific gK peptides that block gK binding to SPP

Using the 33 overlapping peptides, we identified a 15mer gK peptide (RASPLHRCIYAVRPTG) that blocked gK binding to SPP, which we refer to as gK4. To increase gK4 penetration into cells *in vitro* and *in vivo*, we synthesized a peptide containing the HIV Tat peptide (YGRKKRRQRRR). HIV Tat peptide was used as control. gK4 peptide containing HIV Tat and HIV Tat were custom synthesized by GenScript (Piscataway, NJ). To increase the stability of each peptide, they were synthesized in the amino acid "D" form. The sequence of peptide dTat-gK4 is YGRKKRRQRRRASPLHRCIYAVRPTG and the sequence of control dTat peptide is YGRKKRRQRRR. The peptide purity was greater than 95%. Each peptide was dissolved in 1xPBS at a concentration of 4 mg/ml. Ten µg/ml of dTat-gK4 peptide is approximately 3.1 µM, while 10 µg/ml of dTat peptide is approximately 6.4 µM.

## Co-Immunoprecipitation and peptide blocking

For co-immunoprecipitation, HeLa cells were co-transfected with the HA-SPP plasmid and indicated Flag-tagged gK or gK fragment plasmids as we described previously [9]. Cells were harvested at 48 hr post transfection. Cells were lysed in 1x lysis buffer (25 mM Tris PH7.2, 150 mM NaCl, 5% glycerol, 1 mM EDTA, 1% NP-40, with complete protease inhibitor cocktail from Sigma (St. Louis, MO). Lysates were precleared by incubating with control agarose beads for 1 hr at 4˚C. Co-immunoprecipitations were performed at 4˚C unless otherwise indicated. Cell lysate supernatants were incubated with anti-Flag antibody (Sigma St. Louis, MO) at 4˚C overnight. Each peptide was added to the supernatant at a concentration of 5 μg/ml and the protein A/G beads were then added to the supernatant and incubated for another 3 hr at 4˚C. The supernatant was removed, and the beads were washed three times with wash buffer (25 mM Tris, 150 mM NaCl, 5% glycerol, 1 mM EDTA, 0.5% NP-40, and protease inhibitor cocktail). Bound protein was eluted from the washed beads by adding 40 μl of 2× LDS loading buffer with 5% β-mercaptoethanol at 100˚C for 10 min. Protein levels were measured by western Blotting as we described before [14].

## Immunostaining

Vero cells were seeded in LabTek 4-chamber slides (Corning, Big Flats, NY) and infected with VC1 virus (1 PFU/cell for 24 hr) and dTat-gK4 or dTat control peptide were added to culture medium at a concentration of 20 μg/ml. Cells were fixed with 4% paraformaldehyde, washed with 1× PBS, and then permeabilized with 0.3% Triton X-100 in PBS. The cells were then blocked using 1× sea blocker (Thermo Scientific, Rockford, IL) for 1 hr at 25˚C, washed 2 times with 1× PBS, and incubated with anti-V5 (Bethyl Laboratories, Montgomery, TX; catalog number A190-119A) to detect gK and anti-GM130 (Abcam, Cambridge, MA; catalog number ab52649) antibodies. Cells were then washed three times in PBS, and incubated with chicken anti-goat AlexaFluor 488 (for gK) (Life Technologies, Carlsbad, CA) and chicken anti-rabbit AlexaFluor 594 (for GM130) (Life Technologies, Carlsbad, CA) for 2 hr at 25˚C. Slides were washed 3 times with 1x PBS and mounted with Prolong Gold (Invitrogen, Carlsbad, CA). The fluorophores were imaged in separate channels by confocal microscopy using a Leica SP5-X confocal microscope, image acquisition, and data analysis system (Leica Microsystems, Buffalo Grove, IL).

## Fluorescent-Activated Cell Sorting (FACS)

Vero cells were infected with HSV-1 GFP virus at 0.1 PFU/cell for 24 hr in the presence of different concentrations of dTat-gK4 or dTat control peptide. At 24 hr PI, infected cells were harvested by trypsinization and centrifugation. Cells were then washed in cold 1×PBS and fixed in 4% paraformaldehyde. The GFP positive cell population was measured by flow cytometry using Sony SA3800.

**Ocular infection.** Mice were infected ocularly in both eyes with HSV-1 strain McKrae. McKrae was suspended in 2 μl of tissue culture media and administered as an eye drop without prior corneal scarification. C57BL/6 mice were infected with $2 \times 10^5$ PFU per eye, while BALB/c mice were infected with $1 \times 10^5$ PFU per eye as we described previously [13].

## Ocular application of dTat-gK4 peptide as an eye drop

Mice received 20 μg of the dTat or dTat-gK4 peptides as an eye drop in 5 μl of 1× PBS before ocular infection and at 1, 2, 3, 4, and 5 days PI. Peptides were administered twice daily.

## Titration of viruses in tears

Tear films were collected from both eyes of ocularly infected mice from days 1 to 6 PI, using a cotton applicator. Each swab was placed in 1 ml of tissue culture medium and stored at −80˚C until processing. The amount of virus in the medium was determined by a standard plaque assay using RS cells as we described previously [7,20].

## HSV-1 induced eye disease

The severity of blepharitis and corneal scarring was assessed by examination using a slit-lamp biomicroscope. The examination was carried out by investigators blinded to the treatment regimen of the mice and scored according to a standard 0 to 4 scale (0 = no disease, 1 = 25%, 2 = 50%, 3 = 75%, and 4 = 100% staining or involvement). Eyes were examined for blepharitis on peak day of 4 and thereafter blepharitis declines in the eyes of infected mice. Corneal involvement was scored on days 14 and 28 for corneal scarring as we described previously [3].

## RNA extraction from TG and cornea

Corneas and TG from infected mice were collected on day 4 or day 28 after infection. Isolated tissues were immersed in TRIzol reagent (Invitrogen, Carlsbad, CA) and stored at −80˚C until they were processed. The cornea or TG from each animal was processed for RNA extraction as we described previously [21,53]. Isolated total RNA was reverse transcribed with random hexamer primers and murine leukemia virus reverse transcriptase provided in High Capacity cDNA reverse transcription kit (Applied Biosystems, Foster City, CA) according to the manufacturer's recommendations.

## Gene expression analy

**ses**Sequences of the gB (UL27), gK (UL53), UL20, ICP0 (IE110), and LAT custom-made TaqMan primer sets used in this study were as follows: for gB, 5′-AACGCGACGCA CATC AAG-3′ (forward), 5′-CTGGTACGCGATCAGAAAGC-3′ (reverse), and the probe FAM-CAGCCGCAGTACTACC-3′ (where FAM is 6-carboxyfluorescein); for gK, 5′-GGCCACC TACCTCTTGAACTAC-3′′ (forward), 5′-CAGGCGGGTAATTTTCGTGTAG-3′ (reverse), and the probe 5′-FAM-CAGGCCGCATCGTATC-3′; for UL20, 5′-CCATCGTCGGCTAC TACGTTAC-3′(forward), 5′-CGATCCCTCTTGATGTTAACGTACA-3′ (reverse), and probe 5′-FAM-CCCGCACCGCCCAC-3′; for ICP0, 5′CGGACACGGAACTGTTCGA-3′ (forward), 5′-CGCCCCCGCAACTG-3′ (reverse), and the probe 5′-FAM-CCCCATCCACGCCCTG-3′; and for LAT, 5'-GGGTGGGCTCGTGTTACAG-3' (forward), 5'-GGACGGGTAAGTAACA GAGTCTCTA-3' (reverse), and the probe, 5'- FAM-ACACCAGCCCGTTCTTT-3'.

GAPDH primers (Mm999999.15_g1) were used as an internal control. Other TaqMan primers from Applied Biosystems used in this study were CD4 (Mm00442754_m1), CD8 (Mm01182107_g1), PD-1 (ABI Mm00435532_m1); Tim-3 (Mm00454540_m1); IFNγ (Mm00801778_m1); IFNα2 (Mm00833961_s1); IFNβ1 (Mm00439552_s1); and TNF-α (Mm00443258_m1) as we described previously [22].

Quantitative RT-PCR (qRT-PCR) was performed using a TaqMan gene expression assay kit in 384-well plates on a QuantStudio 5 system (Applied Biosystems, Foster City, CA). Copy numbers for gB, gK, UL20, and ICP0 were calculated using standard curves generated from pAc-gB1 (for gB), pGem-gK1040 (for gK), pcDNA-UL20 (for UL20), pcDNA-ICP0 (for ICP0), and pGEM5317 (for LAT). Briefly, each plasmid DNA template was serially diluted 10-fold so that 1 µl contained from $10^3$ to $10^{11}$ copies of the desired gene and was then subjected to TaqMan PCR with the same set of primers as the test samples. Copy number of each

reaction product was determined by comparing the normalized threshold cycle ($C_T$) of each sample to the threshold cycle of the standard curve. For fold change of expression analysis, the $2^{-\Delta\Delta CT}$ method was used to calculate fold change in gene expression compared to expression in uninfected controls.

## Statistical analysis

The Student $t$ test, Fisher exact test, and Mann Whitney test were performed using the computer program Instat (GraphPad, San Diego, CA). Results were considered statistically significant when the $P$ value was $<0.05$.

## Author Contributions

**Conceptualization:** Shaohui Wang, Homayon Ghiasi.

**Data curation:** Shaohui Wang, Ujjaldeep Jaggi, Jack Yu.

**Formal analysis:** Shaohui Wang, Homayon Ghiasi.

**Funding acquisition:** Homayon Ghiasi.

**Investigation:** Shaohui Wang, Homayon Ghiasi.

**Methodology:** Shaohui Wang.

**Project administration:** Homayon Ghiasi.

**Resources:** Homayon Ghiasi.

**Software:** Shaohui Wang.

**Supervision:** Homayon Ghiasi.

**Validation:** Shaohui Wang.

**Visualization:** Shaohui Wang, Homayon Ghiasi.

**Writing – original draft:** Homayon Ghiasi.

**Writing – review & editing:** Shaohui Wang, Homayon Ghiasi.

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
