## [Decision Letter · Decision Letter 0]

18 Apr 2021

Dear Dr. Ghiasi,

Thank you very much for submitting your manuscript "Blocking HSV-1 glycoprotein K binding to signal peptide peptidase reduces virus infectivity in vitro and in vivo" (PPATHOGENS-D-21-00605) for consideration at PLOS Pathogens. As with all papers peer reviewed by the journal, your manuscript was reviewed by members of the editorial board and by several independent peer reviewers. Based on the reviewer reports, we regret to inform you that we will not be pursuing this manuscript for publication at PLOS Pathogens.

The reviewers appreciated the importance of your work, but one had significant concerns with the controls used for many of the experiments. Because of the nature of these experiments, this submission is being rejected to allow you the time necessary to incorporate the suggested controls into your experimental approach, should you wish to re-submit the paper to PLoS Pathogens.

The reviews are attached below this email, and we hope you will find them helpful if you decide to revise the manuscript for submission elsewhere. We are sorry that we cannot be more positive on this occasion. We very much appreciate your wish to present your work in one of PLOS's Open Access publications. 

Thank you for your support, and we hope that you will consider PLOS Pathogens for other submissions in the future.

Sincerely,

Robet Kalejta,

Associate Editor,

PLOS Pathogens

Blossom Damania

Section Editor

PLOS Pathogens

Kasturi Haldar

Editor-in-Chief

PLOS Pathogens

orcid.org/0000-0001-5065-158X

Michael Malim

Editor-in-Chief

PLOS Pathogens

orcid.org/0000-0002-7699-2064

Reviewer's Responses to Questions

**Part I - Summary**

Reviewer #1: Summary: The Ghiasi laboratory has reported previously in a number of publications on HSV-1 glycoprotein K (gK) vis-a-vis its role in the enhancement of virus-induced corneal disease. The mechanism of action of this phenomenon involves the binding of gK to signal peptide pepsidase (SPP). In the present investigation, the authors extended these studies to determine if blocking gK binding to SPP would impact virus infectivity and pathogenicity of corneal disease development in a therapeutic fashion. A series of experiments whose purpose was to fine map the gK binding site to SPP succeeded in identifying a 15 amino acid sequence within the gK molecule that binds to SPP that was designated dTat-gK4 (the 4th 15-mer peptide out of 33 15-mer peptides examined for binding to SPP). Importantly, dTat-gK4 reduced the infectivity of three different strains of HSV-1 that included the virulent McKrae strain of virus. This initial finding allowed for the performance of a subsequent series of experiments involving both BALB/c and C57BL/6 mice that demonstrated the potential therapeutic use of dTat-gK4 when administered to the ocular surface as eye drops. dTat-gK4-treated animals when compared with untreated control animals showed reduced corneal virus replication, increased survival, decreased eye disease (corneal scarring), reduced latency within trigeminal ganglia and a reduction in exhaustion markers of latently infected ganglia. The authors conclude that blocking gK binding to SPP using a gK peptide in the clinical setting could represent a new and effective therapeutic approach to manage HSV-1 ocular disease as well as other HSV-1-associated infections. 

Review: This is a well written manuscript that nicely summarizes a large body of impressive and comprehensive investigations that provides new and exciting evidence for use of a gK peptide to manage HSV-1 ocular disease. The multiple experimental approaches lead to convincing data that support the overall premise of the investigation regarding the possible clinical therapeutic value of a gK peptide. While already a strong investigation, attention to some concerns might serve to further strengthen the manuscript for readers. 

1. The McKrae strain of HSV-1 has been used by many laboratories to investigate the pathogenesis of corneal HSV-1 disease and most investigators therefore are familiar with its use in this area of investigation. The present study uses two additional viruses, however, that are designated HSV-1 GFP and HSV-1 VC1. On what virus strain background was HSV-1 GFP constructed? Some additional information on HSV-1 VC1 would also be helpful including a reference of its origin. 

2. Because HSV-1 strains other than the McKrae strain require corneal scarification for their effective use in corneal disease investigations, the authors should emphasize in the Discussion section that corneal scarification is not used in the present investigation, an experimental approach that heightens the clinical significance of their work. 

3. Although data showing the fine mapping of the gK binding region to SPP are convincing, to what extent does the secondary structure of gK also contribute to efficient binding of these two molecules?

4. What is the rationale for measuring blepharitis at 4 days after ocular infection?

5. In vivo studies use an experimental plan whereby dTAT-gK4 is applied to the cornea prior to HSV-1 infection and thereafter at days 1, 2, 3, 4, and 5. To what extent is this experimental protocol clinically relevant? The findings provide impressive proof-of-principle, but patients with HSV-1 corneal disease would not be treated with this protocol in the clinical setting. The authors might comment how their very important findings might translate to treatment of patients in the clinic.

Reviewer #2: Wang et. al. present work building on several previously published papers regarding the role of HSV-1 gK in the pathogenesis of viral keratitis. In this work, they extend the finding that gK binds to signal peptidase and use a peptide based strategy as a potential therapeutic for the disease. The major finding presented is that treatment with the peptide reduces infection in cell culture and in animal models reduces viral titers, clinical disease severity, and cytokines in the cornea. The work is potentially interesting but there are significant issues with the study design and other matters.

**Part II – Major Issues: Key Experiments Required for Acceptance**

Reviewer #1: No key experiments are required.

Reviewer #2: The most significant weakness in the study design is the lack of a proper control inactive peptide. In order to deliver their gK peptide, they coupled it to the TAT peptide that is known to facilitate transport of materials into cells (not properly cited). They then use the TAT peptide only as their negative control. This is not appropriate because they have an altered gK peptide that is inactive and this peptide coupled to TAT is the appropriate control.

A second issue is the lack of proper referencing. There is a considerable literature on the use of peptides as antivirals and more specifically, as potential treatments for viral keratitis. None of these papers are cited. The authors completely ignored this body of literature and do not place their results in the context of the existing literature.

A third issue is their data indicate the TAT peptide is inactive when other publications have shown it, and some derivatives of TAT, have antiviral activity. This discrepancy is not explained or even addressed.

There is no vehicle only control in the animal experiments and they do not provide the gender of the mice. They should have tested both male and female mice or explained why this is not necessary.

**Part III – Minor Issues: Editorial and Data Presentation Modifications**

Reviewer #1: 1. The first line of the Abstract states "HSV glycoprotein K (gK) is an essential herpes gene", a statement that is not correct. The glycoprotein is the product of a gene but not the gene itself. This should be corrected.

Reviewer #2: It is absolutely essential that all peptide concentrations should be given in micromolar not micrograms/volume. Using the weight makes it difficult to compare the efficacy of their peptide with other antivirals or antiviral peptides. They do not discuss the efficacy of their peptide in the context of the efficacy of other antiviral peptides.

The presentation of the data in Figure 3 is potentially misleading to readers. At 50 micrograms per ml. concentration the reduction is less than 1 log. By plotting the Y axis to extend only over a 1 log span makes the differences look greater than they are so unless readers pay attention to the scale they will think the peptide is working well when if fact it is similar to other antiviral peptides in the literature. Again, all concentrations should be micromolar.

They do not provide an EC50 value as is commonly done with antivirals.

In the second sentence of the introduction the authors make the statement "...Although the HSV-1 genes involved in eye disease is not yet known...." This statement is absolutely incorrect. There are several papers and review articles describing the role of HSV genes in ocular infections. Once again, the existing literature is not properly cited.

PLOS authors have the option to publish the peer review history of their article (what does this mean?). If published, this will include your full peer review and any attached files.

Reviewer #1: No

Reviewer #2: No

---

## [Decision Letter · Decision Letter 1]

22 Jul 2021

Dear Dr. Ghiasi,

Thank you very much for submitting your manuscript "Blocking HSV-1 glycoprotein K binding to signal peptide peptidase reduces virus infectivity in vitro and in vivo" for consideration at PLOS Pathogens. As with all papers reviewed by the journal, your manuscript was reviewed by members of the editorial board and by several independent reviewers. The reviewers appreciated the attention to an important topic. Based on the reviews, we are likely to accept this manuscript for publication, providing that you modify the manuscript according to the review recommendations.

Sincerely,

Zhen Lin

Associate Editor

PLOS Pathogens

Blossom Damania

Section Editor

PLOS Pathogens

Kasturi Haldar

Editor-in-Chief

PLOS Pathogens

orcid.org/0000-0001-5065-158X

Michael Malim

Editor-in-Chief

PLOS Pathogens

orcid.org/0000-0002-7699-2064

Reviewer Comments (if any, and for reference):

Reviewer's Responses to Questions

**Part I - Summary**

Reviewer #1: This is a revised manuscript that provides new and important information on the use of a gK peptide to manage HSV-1 corneal disease, findings that may have significant clinical relevance. The manuscript is well written, and the results and conclusions are convincing and compelling. All concerns and questions raised by the previous reviewers have been answered thoughtfully and thoroughly with appropriate revisions throughout the manuscript. No additional weaknesses or concerns have been identified.

Reviewer #3: Wang et. al. present an intriguing study on the use of a novel peptide to decrease HSV-1 disease in a murine model of HSV-1 eye disease. The authors build on their previous extensive work on gK and its binding to SPP to map the region of gK responsible for SPP binding. They then synthesize a novel molecule that contains the gK 15mer (gK4) responsible for binding to SPP bound to HIV TAT to improve uptake in cells. They then go on to demonstrate that this molecule inhibits HSV-1 replication in vitro and in vivo and improves clinical HSV disease scores. Finally, they present some gene expression data to suggest the mechanism of disease prevention does not impact the host immune response to the virus. While there are doubts about the feasibility of using this specific compound in a clinical setting, the mapping of the interaction between gK and SPP and this proof of principle study are highly relevant to the HSV-1 field and highlights a novel mechanism of pathogenesis. Overall, this was a well-designed and executed study, and the manuscript is easy to follow.

**Part II – Major Issues: Key Experiments Required for Acceptance**

Reviewer #1: None

Reviewer #3: Major Concerns:

-I did not identify any serious methodological flaws. No obvious controls are lacking (save for one discussed below). For the most part their conclusions are supported by their data apart from a few exceptions below. I do have some suggestions to improve the presentation of the data and ensure proper statistical analysis which are discussed below. However, I do not believe they will significantly alter the results or conclusions.

**Part III – Minor Issues: Editorial and Data Presentation Modifications**

Reviewer #1: None

Reviewer #3: Minor concerns

-Abstract line 23: The use of “fine mapped” is non-specific and the authors have a definitive length of a 15mer so no reason to be non-specific. Recommend removing the modifier “fine.”

-Intro line 45. I agree with reviewer #2’s comment that to say that the HSV genes are not known seems to discount a large body of literature. Perhaps something like “While the precise viral pathogenicity determinant in ocular disease remains to be elucidated…”

-Intro line 65: This sentence starts with “We also asked…” but then there is no question. Perhaps the authors mean “We also sought to determine?” There is a similar issue in the results on line 169.

-Intro line 72: The sentence that reads “Thus, we have identified an effective therapeutic target to reduce HSV-1 primary infection, latency, and HSV-1-induced eye disease .” is problematic. First, it feels like it should go into the discussion. Second, it is not supported by the data in the manuscript. This study is entirely done using a mouse model that, while is good as far as mouse models go, does not perfectly translate to human infection. No primary human corneal cells are used here nor other human tissues. Not to mention that using eye drops at the time of infection is entirely not feasible in humans. This sentence needs to be toned down. The sentence immediately following is entirely appropriate.

-Figure 3: Why did the authors stop at 50ug/mL? Would increasing the dose further reduce infectivity? Some type of viability control should be provided. Is this compound toxic and thus killing the cells? Why did the authors leave the viral inoculum on for the whole time? I think that a 2 hour incubation would be sufficient. Where are the error bars or individual data points in this figure? Are there stats done here?

-Figure 4: The use of error bars (SEM or Sd) is less than ideal. Would be better to show individual data points as is done in Figure 6B.

-Figure 5: These data need to be quantifed. The authors make the statement “more cells were positive than in cells incubated with dTat-gK4,” but the reader is left with only 1 image. It sounds as though the authors have already quantified these data, but it should be displayed as such.

-Figure 6B: The statistical test done here is a Student’s t-test according to the methods, but these data do not look normally distributed and there are not more than 30 samples per group. Therefore, either some test of normality should be performed, or a non-parametric test should be performed.

-Figure 6C: This is truly a minor point. The way these data are displayed gives the impression that this is a Kaplan-Meyer survival curve, but the authors did not do a survival analysis. They performed a Fisher’s exact test at a specific time point (which I think is the correct test here). Alternatively, the data could simply be displayed in a table. The way it is currently displayed makes the reader think that the mice treated with control actually died FASTER…but again, that wasn’t the test that was done.

-Results line 197: The authors mention measuring mRNA for various HSV proteins (gK, gB, etc.) I believe since they are looking at gene expression they should use the gene designations (UL20 etc.)

-Figure 8: The use of bar graphs with error bars is not very transparent and can hide obvious outliers. This would better be displayed like the data in figure 6B.

-Discussion: There is too much restating of findings early on, see lines 315 and the rest of that paragraph.

-Discussion line 363: what is meant by “virus load?” Input virus? Titer of inoculation virus? DNA? PFU? Please clarify.

-Discussion line 380: the comment “these results suggest that T cell responses and viral clearance are enhanced” is not supported by the data. The authors did not directly measure T cell responses in this study. Gene expression profiling on bulk tissue does not isolate T cell responses. They did not measure whether T cells were more or less responsive to viral antigen. They also did not demonstrate that gK4 treatment enhanced viral clearance. My understanding is that the proposed mechanisms is that gK4 inhibits viral replication at the site of inoculation by interfering with gK binding to SPP. This has nothing to do with cells of the innate or adaptive immune system sensing and/or clearing the virus. This sentence needs to be removed.

PLOS authors have the option to publish the peer review history of their article (what does this mean?). If published, this will include your full peer review and any attached files.

Reviewer #1: No

Reviewer #3: No

Figure Files:

Data Requirements:

Reproducibility:

References:

---

## [Editor Report · Decision Letter 2]

28 Jul 2021

Dear Dr. Ghiasi,

We are pleased to inform you that your manuscript 'Blocking HSV-1 glycoprotein K binding to signal peptide peptidase reduces virus infectivity in vitro and in vivo' has been provisionally accepted for publication in PLOS Pathogens.

Best regards,

Zhen Lin

Associate Editor

PLOS Pathogens

Blossom Damania

Section Editor

PLOS Pathogens

Kasturi Haldar

Editor-in-Chief

PLOS Pathogens

orcid.org/0000-0001-5065-158X

Michael Malim

Editor-in-Chief

PLOS Pathogens

orcid.org/0000-0002-7699-2064
---

## [Editor Report · Acceptance letter]

3 Aug 2021

Dear Dr. Ghiasi,

We are delighted to inform you that your manuscript, "Blocking HSV-1 glycoprotein K binding to signal peptide peptidase reduces virus infectivity in vitro and in vivo," has been formally accepted for publication in PLOS Pathogens.

Best regards,

Kasturi Haldar

Editor-in-Chief

PLOS Pathogens

orcid.org/0000-0001-5065-158X

Michael Malim

Editor-in-Chief

PLOS Pathogens

orcid.org/0000-0002-7699-2064